# Postharvest bacterial succession on cut flowers and vase water

Yen-Hua Chen[1¤], William B. Miller[1], Anthony Hay[2]*

1 Department of Horticulture, School of Integrative Plant Science, College of Agriculture and Life Sciences, Cornell University, Ithaca, New York, United States of America, 2 Department of Microbiology, College of Agriculture and Life Sciences, Cornell University, Ithaca, New York, United States of America

¤ Current address: Taichung District of Agricultural Research and Extension Station, Ministry of Agriculture, Dacun Township, Changhua County, Taiwan

* agh5@cornell.edu

**Data Availability Statement:** All relevant data are within the paper and its Supporting information files.

**Funding:** The author(s) received no specific funding for this work.

## Abstract

In cut flowers, xylem occlusion or blockage by bacteria negatively affects water balance and postharvest quality. Many studies have used culture-based methods to examine bacterial populations in vase water and their effects on flower longevity. It is still unclear if and how bacterial communities at the 16S rRNA gene (16S) level change during the vase period and how such change might correlate with postharvest longevity. This study compared the sequences of 16S amplicons from 4 different types of flowers and their vase water over the course of 7 days (*Rosa spp.*, *Gerbera jamesonii*, and two *Lilium* varieties). The relative abundance of plant chloroplast and mitochondria 16S decreased significantly over the course 7 days in all 4 flowers as bacterial diversity increased. Richness and evenness of the bacterial communities increased over time, as did the number of rare taxa and phylogenetic diversity. Bacterial communities varied with time, as well as by flower source, types, and sample location (water, stem surface, whole stem). Some taxa, such as *Enterobacteriacea* and *Bradyhizobiaceae* decreased significantly over time while others such as *Pseudomonas spp.* increased. For example, *Pseudomonas veronii*, implicated in soft rot of calla lily, increased in both whole stem samples and water samples from *Gerbera jamesonii*. *Erwinia spp.*, which includes plant pathogenic species, also increased in water samples. This work highlights the dynamic and complex nature of bacterial succession in the flower vase eco-system. More work is needed to understand if and how bacterial community structure can be managed to improve cut flower vase life.

## Introduction

In cut flowers, the maintenance of open xylem vessels to facilitate long-term water uptake is a critical aspect of maintaining flower vase life. Xylem occlusion or blockage from bacterial proliferation in and around the stem is a common phenomenon with cut flowers. In previous research, $> 10^7$ colony forming units (cfu)·mL$^{-1}$ of bacteria caused lower hydraulic conductance [1, 2], shorter flower life [3, 4], less fresh weight [5, 6], and inferior flower quality [7, 8].

**Competing interests:** The authors have declared that no competing interests exist.

Bacteria affecting postharvest quality were mainly from the stems, water, or soil [9]. Put (1990) indicated that different locations on the flowers had different dominant microorganisms, likely due to distance from the soil [9]. Taxa of bacteria previously reported to affect the postharvest quality of cut flowers include *Pseudomonas solanacearum* in the vase solution of carnations [10], *Pseudomonas aeruginosa* from roses [2], and *Enterobacter spp.* and *Bacillus spp.* for chrysanthemum and gerbera, respectively [9].

Previously, the identification of flower-associated bacteria involved plating and purification on agar followed by the use of commercial systems such as OXI/FERM [11]. The same methods were used to identify bacteria from *Gerbera jamesonii* [12]. The accuracy of OXI/FERM systems, however, has been reported to be poor with nonfermentative Gram-negative bacteria [13]. More recently, 16S rRNA gene (16S) sequencing has proven to be a useful and well-developed technique for identifying bacterial species. Carlson et al. (2015) applied 16S analysis for bacterial identification, but only for isolated pure colonies that could not be identified by BIO-LOG Microlog 3 (BIOLOG, Inc., CA, USA) [14]. The identified microorganisms from cut *Zinnia* included *Pseudomonas fulva*, *Serratia ficaria*, *Rhizobium radiobacter*, *Chryseobacterium spp.*, *Pantoea ananatis*, *Bacillus pumilus*, *Chryseobacterium daejeonense*, and *Brevundimonas spp.* [14]. In many ecosystems, however, cultured bacteria represent less than 2% of the bacterial world [15]. The identification of bacteria based on pure cultures may therefore miss the other > 98% of bacterial taxa.

Since bacteria have such significant effects on postharvest quality, it is important to understand which bacteria are present when, in order to learn more about their roles in postharvest physiology. Previous culture-based studies revealed that certain genera of bacteria tend to be present on different flowers, or only analyzing the bacterial community on the stem end only, and just one time during vase period [16]. It is unclear if and how the composition of bacterial communities' changes during the vase period. In this project, we aimed to begin answering these questions by analyzing the 16S sequences amplified from DNA extracted from water and flower stems. The research hypothesis was that bacterial communities would change by flower source, type, and location (water vs. stem) over time. Therefore, we compared the bacterial communities between flowers as well as the changes of bacterial communities during the vase period.

## Material and methods

### Plant material

*Lilium* 'Sorbonne' stems with four buds were harvested from the greenhouse at Cornell University. The other cut flowers, unknown cultivar of Asiatic hybrid lily (*Lilium spp.*), rose (*Rosa spp.*) and gerbera daisy (*Gerbera jamesonii*) were purchased from a local grocery store and originated from unidentified farms in South America. The flowers were shipped in cardboard boxes to the local store and dry stored at 4 °C before arrival in the lab. Stems were trimmed to 60–70 cm length with leaves removed from the bottom 15 cm. Cutting tools were sanitized with a solution of 70% alcohol and 10% bleach between each stem.

### Sampling approach

Vases were cleaned with 10% bleach solution and rinsed with reverse osmosis water twice and Milli-Q water (Milli-Q® Direct Water Purification System) twice. After cleaning, we placed vases upside down to air dry before using. Tap water from the Kenneth Post laboratory at Cornell University was used as vase water; each vase received 600 mL tap water with 12 stems per vase. Water volume was checked daily and refilled with Milli-Q water to 400 mL in the vase on day 3 if vase water was less than 400 mL. We used a vase with 600 mL tap water without any

flowers as a control. A 15 mL water aliquot was sampled from each vase holding flowers on day 1, 3, and 7, and the control vase on day 0, 1, 3, 7. The sampling time of day 0 was October 31, 2020. Sterile 15 mL centrifuge tubes were used for water sampling. As for stem samples, 3 stems of each species were taken from each vase on day 0, 1, 3, and 7. Stem samples of day 0 were collected before being placed into vase water. The 5 cm basal end of the stems were cut into 1 cm segments for bead beating (to sample internal and external bacteria associated with the stems) and 4 cm segments for swabbing stem surfaces (to sample external bacteria only). Stem segments from each sampling were wrapped in aluminum foil and stored at a -20 $^{\circ}$C before DNA extraction.

## DNA sampling extraction, PCR amplification, and sequencing

For water samples, 2 mL were filtered by Sterivex™ pressure-driven filter (Millipore Sigma Inc., Massachusetts, USA) using a sterile syringe following the instructions of the DNeasy Power-Water Sterivex kit (QIAGEN, Hilden, Germany). To sample external stem bacteria, we used a sterile cotton swab to wipe the surface of a 4 cm basal stem segment back and forth (3 stems/ flower type). The head of the cotton swab was added to a PowerBead Tube provided in the DNeasy PowerLyzer PowerSoil (QIAGEN, Hilden, Germany). The 1 cm stem segment was cut longitudinally into two pieces by a sterile razor which was cleaned with 10% bleach and sanitized with a 70% alcohol solution followed by Bunsen burner fire each time it was used. Then, we put stem segments into a PowerBead Tube provided in the DNeasy PowerLyzer PowerSoil Kit and followed the manufacturer's instructions to extract DNA. The extracted DNA samples were stored at -20˚C until further analysis.

We used agarose gel electrophoresis to check the extracted DNA quality and PicoGreen quantification kit to normalize DNA concentration. The V4 region of the bacterial 16S rRNA gene was amplified from the DNA extracts of both water and stems. The PCR program was set for 95˚C for 2 min, then 29 cycles at 95 $^{\circ}$C for 20 s, 55 $^{\circ}$C for 15 s, and 68 $^{\circ}$C for 30 s with a final extension of 68 $^{\circ}$C for 5 min followed by 4 $^{\circ}$C ending. The AccuPrime™ *Pfx* DNA polymerase (Invitrogen Corporation, Carlsbad, CA) was used for the PCR amplification with primer of 515F 5'-GTGYCAGCMGCCGCGGTAA-3' and 806 R 5'-GGACTACNVGGGTWTCTAAT-3'. For post-sequencing sample identification, each primer also included a "barcode" which was a unique 8-nucleotide with the primers [17]. Amplicon concentrations were normalized using a SequalPrep™ Normalization Plate Kit (Thermo Fisher Scientific Inc., Waltham, Massachusetts) and pooled together for sequencing. Paired-end sequencing (2 x 250 bp) was conducted on an Illumina MiSeq sequencer at the Cornell Sequencing Center, Ithaca NY, USA.

## Statistical analysis

The 16S rRNA sequencing results and statistical analysis were processed using QIITA [18] without rarefaction and based on closed-reference of operational taxonomic units (OTU) picking process based on 97% 16S similarity of Greengenes database version 3_8–97. Chloroplast and mitochondria reads as well as features with fewer than 100 reads across all samples were filtered out before bacterial community analysis. The Shannon, Chao 1, and Faith's phylogenetic indices of alpha diversity were calculated in QIITA and plotted with JMP (SAS co.) version 15.1 with Kruskal-Wallis statistic test. The heatmaps based on compete linkage clustering and the Principal Component Analyses were generated using ClustVis [19]. Beta diversity between samples was also calculated using generalized UniFrac with 0.5 alpha control and significance was determined via PERMANOVA [20]. A volcano plot was generated using

VolcaNoseR [21]. The ClustVis and VolcaNoseR plots were based on the relative abundance of the top 20 most abundant OTUs that accounted for > 95% of total reads.

## Results

### The dynamics of chloroplast and mitochondrial small subunit (SSU) amplicons during the vase period

Chloroplast and mitochondrial sequences are often removed from bacterial 16S community analyses, but in our case, where only one flower type was present in each sample, we thought it would be informative to understand the dynamics of chloroplast and mitochondrial 16S amplicons so the data was included to shed insights on flower physiology. Fig 1 shows the relative abundance of a) chloroplast and b) mitochondria 16S sequences (% of all amplicons) in whole stem segments for each of the flower types. In all 4 flower types, The initial relative abundance of chloroplast sequences ranged from 25–85% (Fig 1A). Chloroplast relative abundance decreased significantly over seven days for all samples (Pt0 vs t7 <0.001), yielding fold changes ranging from 3.6 to 32.8. The relative abundance of mitochondria 16S in whole stem segments showed a similar pattern (Fig 1B). The initial relative abundance of mitochondria sequences ranged from 6% to 30% of total reads and decreased significantly over time (Fig 1).

### The bacterial diversity of flowers and vase water by source and cultivar

Alpha diversity analyses at the OTU level (Shannon entropy, Chao 1, and Faith's phylogenetic diversity) showed similar trends for all four flower types. Representative examples for *Lilium* 'Sorbonne' and *Gerbera* are presented in Fig 2 (See S1 Fig for the results of *Rosa spp.* and *Lilium*-As).

 Shannon entropy is an estimator of species richness and evenness of bacterial communities with more weight on richness. The OTU richness and evenness of the all the flower bacterial communities (water, stem surface, whole stem) increased over time. Results from the Chao 1 index, which gives weight to rare species, also increased over time from < 5 to ca. 25. The change in Faith's phylogenetic diversity suggests that increased OTU richness was driven by phylogenetic dissimilarity taxa (Fig 2).

 Given that flower bacterial alpha diversity increased over time, we wanted to know if the changes followed similar trajectories for different flower types. Comparison of bacterial communities between flower types is presented via Principal Component Analysis (PCA) of the 20 most abundant taxa that accounted for ~95% of bacterial reads (Fig 3). The bacterial communities varied significantly by flower source and type. In general flower type had a bigger effect on community composition than time or sample location (Fig 3). The average values for both PCs were different for each flower type, although the three flowers that came from the same vendor occupied a similar location on PC1, with *Lilum*-Sorbonne and rose also showing some overlap on PC2 (Table 1).

 There are two flower sources, *Lilium* 'Sorbonne' was grown in the greenhouse of Cornell University, and the other three flower crops were purchased from the local supermarket. *Gerbera jamesonii* is one of the major cut flowers in the globe. The gerbera flower scape surface is hairy, and the anatomical structure of the scape is dissimilar to the other flower types. Based on Fig 3, Table 1, and S1 Table, the further investigations of *Lilium* 'Sorbonne' and *Gerbera jamesonii* as our representative flowers are present in Figs 4 and 5.

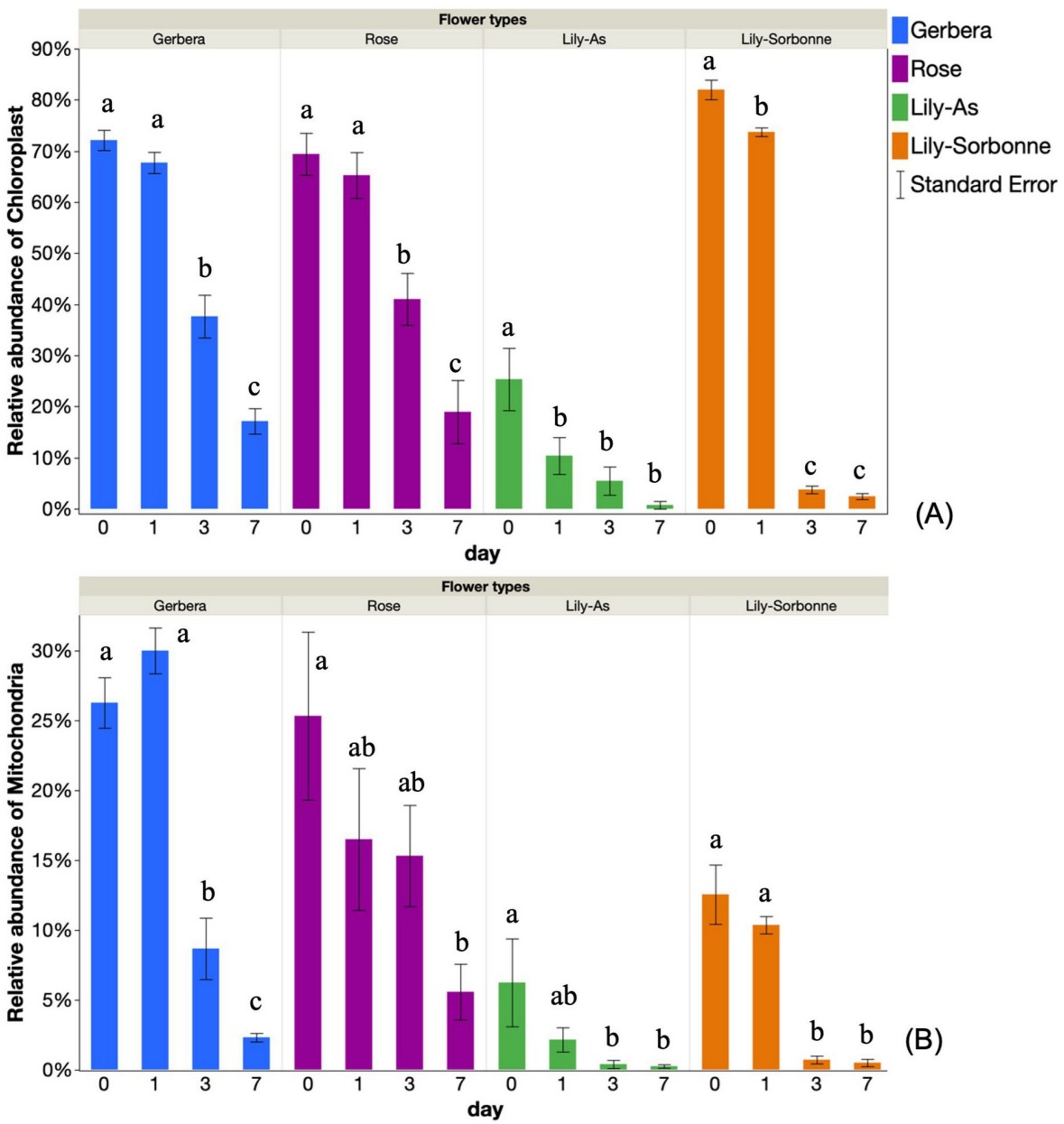

**Fig 1. Relative abundance of chloroplast and mitochondria in whole stem segment over time with different flowers.** (A) Chloroplast (B) Mitochondria. Data are means of 3 replicates. Bars are standard errors (n = 3). The same letter in the same flower type indicated data were not significantly different using one-way analysis of variance (ANOVA), followed with least significant difference test (*p*-value < 0.05).

## Community differences between sampling locations over time for *Lilium* 'Sorbonne' and *Gerbera jamesonii*

Within a given flower type, the generalized Unifrac analyses revealed that bacterial communities differed by sampling location and over time (Table 2). Examples for *Lilium* 'Sorbonne' and *Gerbera jamesonii* are present in Figs 4 and 5. For *Lilium* 'Sorbonne', both factors, "Sample type" and "Day" in the vase affected bacterial compositions, although the interaction of type

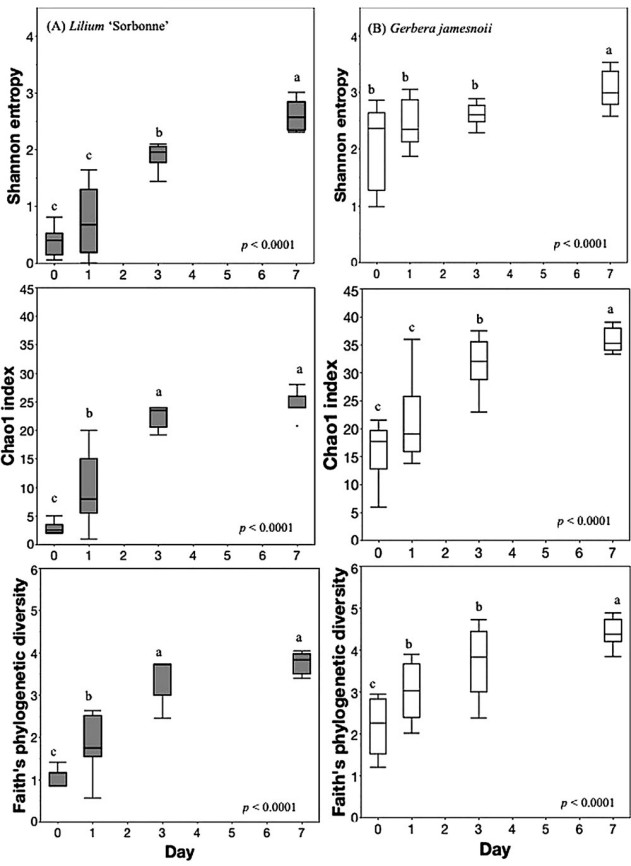

**Fig 2. Shannon entropy, Chao 1, and Faith's phylogenetic diversity of bacterial communities over time.** (A) *Lilium* 'Sorbonne' (B) *Gerbera jamesnoii*. The statistical differences were tested using Wilcoxon/Kruskal-Wallis Test. Different letters represent significance with $p < 0.05$.

and day only bordered on significant ($p = 0.06$). "Day" explained more variation in bacterial compositional change than "Sample type" based on $R^2$ value. Results for rose and *Gerbera* were similar, whereas the interaction of type and day was not significant for *Lilium*-As ($p = 0.14$).

Changes in the bacterial community of *Lilium* 'Sorbonne' over time are shown in Fig 4. The bacterial communities from day 0 and day 1 grouped together and were significantly different from those on day 3 and 7. In addition to a shift in the location of ellipsoids (95% confidence), the size of the ellipsoids also changed, suggesting increased variation within sample type. Although their locations on the PC plot differed from Sorbonne, a similar pattern of community succession was observed for *Gerbera jamesonii*, with days 0 and 1 showing overlap and a distinct shift by day 3 followed by even greater difference by day 7 (Fig 5). The largest shifts in the PCA over time appeared to be driven by changes on the stem surface and whole stem, rather than the water itself as evidence by the overlapping location of water samples for days 3 and 7.

The heatmaps of the most abundant OTUs from *Lilium* 'Sorbonne' (Fig 6) and Gerba (Fig 7) are consistent with results from the PCAs (Figs 4 and 5), with complete linkage clustering confirming that, although initially similar (days 0 and 1), the bacterial communities diverged over time, with further resolution occurring by sample type (stem vs water).

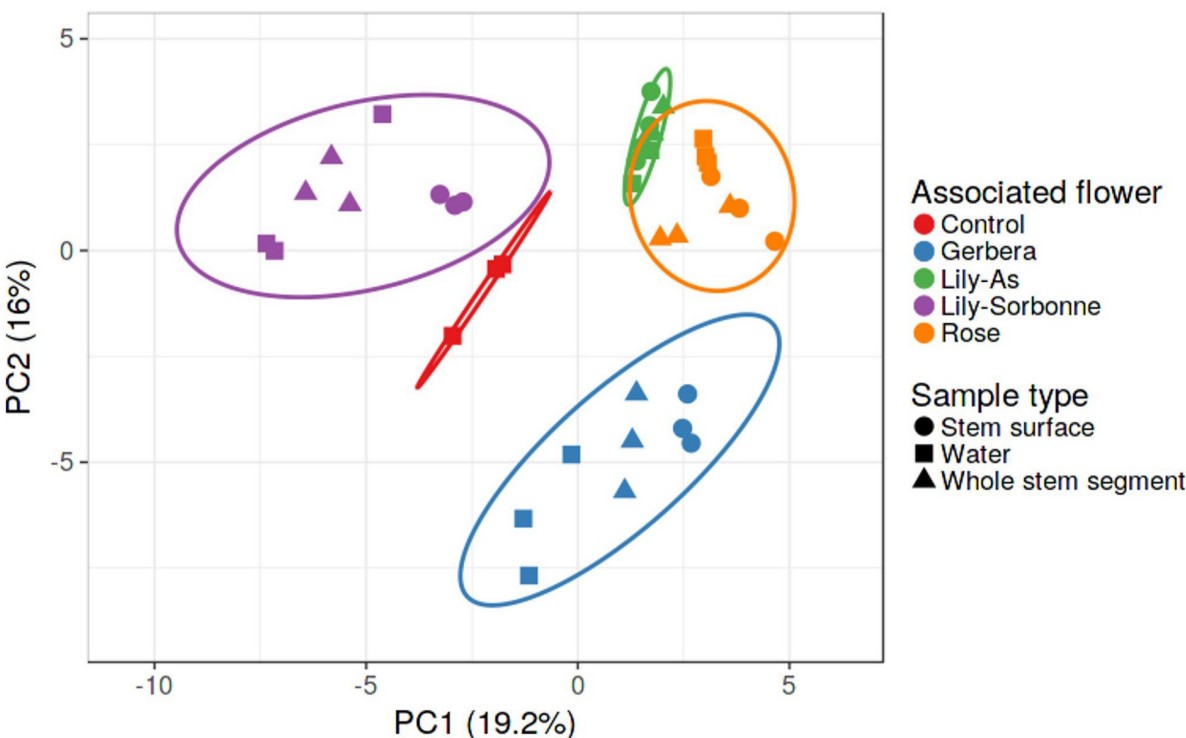

**Fig 3. Principal components analysis (PCA) of bacterial communities from different flowers.** The stem surface and whole stem segments were from days 0, 1, 3, and 7; water samples were from days 1, 3, 7. Control group was vase water only without holding any plants. The PCA is based on the relative abundance of the top 20 OTUs and plotted with ClustVis. Flowers types are presented as different colors, and sample type by different shapes. The ellipses represent 95% confidence intervals. The values in parentheses are the percentages of the total variance explained by each component.

To better understand changes in specific OTUs over time, we constructed a volcano plot for whole stem samples and water samples of *Gerbera jamesonii* (Fig 8). *Planctomyces spp.*, *Sphingomonadaceae*, *Bradyhizobiaceae*, and *Enterobacteriaceae* decreased significantly with $\log_2$ fold changes ranging from 1.5 to 10 on days 3 and 7 compared with day 1. *Pseudomonas veronii*, on the other hand, increased significantly in both whole stem and water samples with $\log_2$ fold changes of 5 to 14 over the same time period. For some taxa, however, changes were

**Table 1. The comparison of principal components in different flower bacterial communities.**

|  | PC1 | PC2 |
|---|---|---|
| *Gerbera jamesonii* | 0.682[a] b[b] | 4.999 a |
| *Rosa spp.* | 3.149 a | -0.798 c |
| *Lilium*-As. | 1.985 a | -1.416 c |
| *Lilium* 'Sorbonne' | -5.133 d | -3.079 d |
| Control group (water only) | -2.049 c | 0.879 b |
| *p*-value | <0.001 | <0.001 |

[a]Data are principal component value for all samples per flower.

[b]Data in columns with the same letter were not significantly difference (*p*-value < 0.05) tested with one-way analysis of variance (ANOVA) followed with least significant difference test.

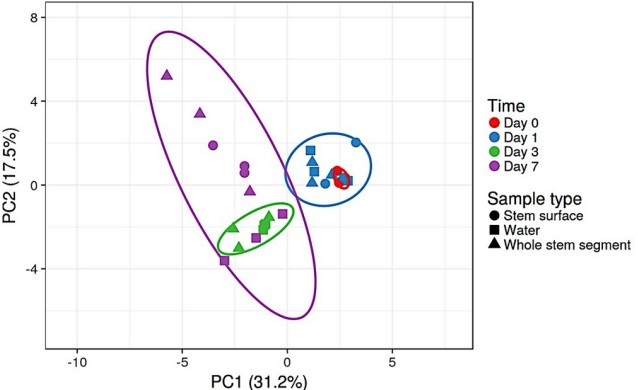

**Fig 4. Principal components analysis (PCA) of bacterial communities over time for *Lilium* 'Sorbonne'.** Water samples included days 1, 3, 7. Stem surface samples and whole stem segment samples included days 0, 1, 3 and 7. The PCA is based on the relative abundance of the top 20 OTUs and plotted using ClustVis. The ellipses represent 95% confidence intervals. The values in parentheses are the percentages of the total variance explained by each component.

only apparent in one sample type. For example, OTUs from *Xanthomonadaceae* increased in the whole stem samples with a range of $\log_2$ fold changes from 4.4 to 6.2 (*p*- value = 0.017), but not in the water samples, whereas *Stenotrophomonas* OTUs only increased in the water samples over time with a range of $\log_2$ fold changes from 3.9 to 4.1 and *p*-value = 0.0019.

## Discussion

After harvest, most cut flowers rapidly proceed through senescence to death, with bacterial xylem occlusion as one of the factors decreasing flower life [22]. Our previous experiments [23] quantified the increase in culturable bacteria in stem segments over time but did not reveal the taxonomic affiliation of those bacteria. More recently, Li et al. (2019) reported a culture independent analysis of bacterial 16S amplicons from *Gerbera jamesonii* vase water, however, they did not analyze amplicons from the stems [24].

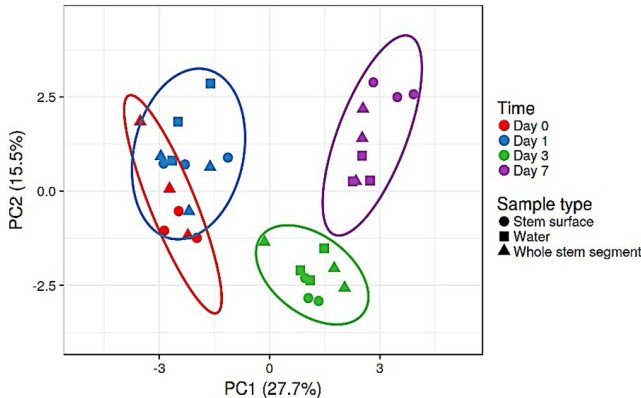

**Fig 5. Principal components analysis (PCA) of bacterial communities in different days in *Gerbera jamesonii*.** Water samples included days 1, 3, 7. Water samples in the Stem surface and whole stem segment samples included days 0, 1, 3 and 7. The PCA is based on the relative abundance of the top 20 OTUs and plotted using ClustVis. Colors represent days of vase period. The ellipses represent 95% confidence intervals. The values in parentheses are the percentages of the total variance explained by each component.

**Table 2. Adonis PERMANOVA (permutational multivariate analysis of variance) test comparing bacterial communities' Unifrac dissimilarity by sample type and days of different flowers.**

| Factors | *Lilium* 'Sorbonne' | | *Gerbera jamesonii* | | *Lilium*-AS | | *Rosa spp.* | |
|---|---|---|---|---|---|---|---|---|
| | $R^2$ | *p*-value | $R^2$ | *p*-value | $R^2$ | *p*-value | $R^2$ | *p*-value |
| **Sample type**[a] | 0.160 | 0.002 | 0.120 | 0.003 | 0.154 | 0.001 | 0.235 | 0.001 |
| **Day**[b] | 0.427 | 0.001 | 0.330 | 0.001 | 0.456 | 0.001 | 0.338 | 0.001 |
| **Sample type * Day** | 0.067 | 0.061 | 0.092 | 0.007 | 0.041 | 0.148 | 0.105 | 0.001 |
| **Residuals** | 0.346 | - | 0.458 | - | 0.349 | - | 0.322 | - |
| **Total** | 1.000 | - | 1.000 | - | 1.000 | - | 1.000 | - |

[a]Sample type includes whole stem segment, stem surface, and water sample.

[b]Day includes day 0, 1, 3, and 7.

[c]Adonis PERMANOVA was processed via QIITA.

This work provides a culture independent analysis of bacterial and plant 16S genes (chloroplast, and mitochondria) in vase water and stem samples from cut flowers over time. Chloroplast and mitochondria are bacteria-derived organelles whose small subunit rRNA gene amplifies with "universal" 16S PCR primers. Chloroplast and mitochondria sequences are

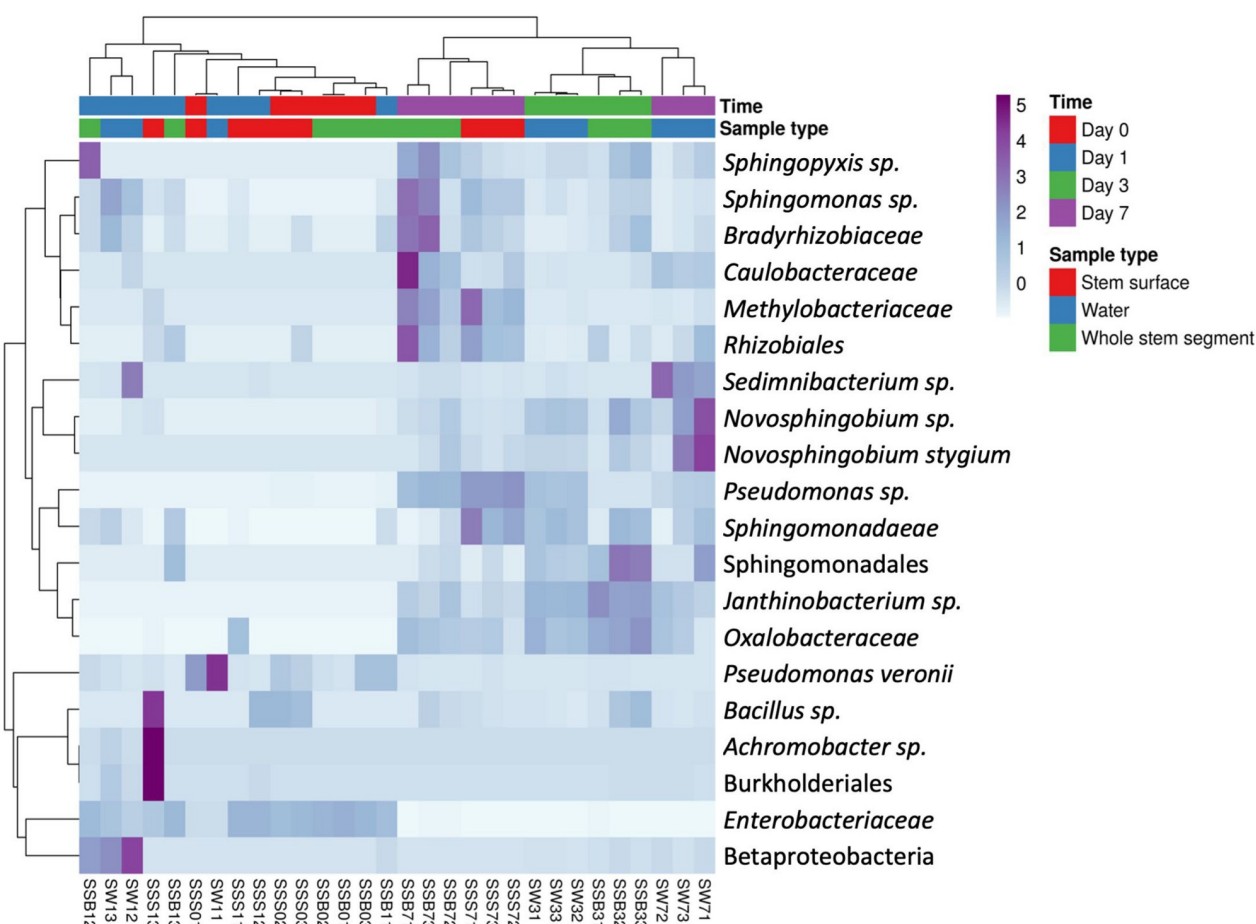

**Fig 6. Heatmap of the top 20 OTUs bacterial taxa in *Lilium* 'Sorbonne'.** Samples are clustered based on the relative abundance of the taxa represent by correlation distance and average linkage. Sample ID: 1st letter: S, Sorbonne; 2nd and 3rd letter: water sample (W), stem surface (SS), whole stem segment (SB); 1st number: day (time); 2nd number: replicate.

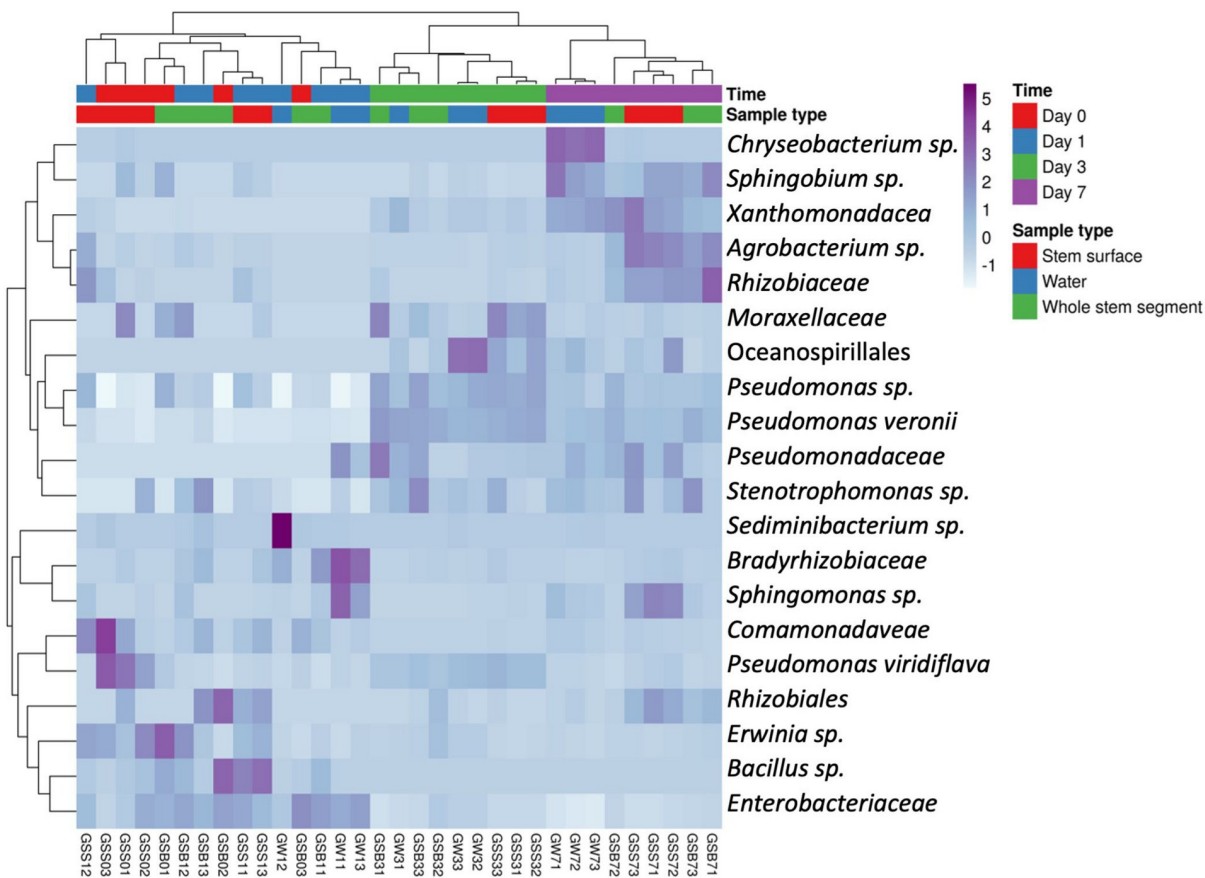

**Fig 7. Heatmap of the top 20 OTUs bacterial taxa in *Gerbera jamesonii*.** Samples are clustered based on the relative abundance of the taxa by correlation distance and average linkage. Sample ID: 1st latter: G, Gerbera; 2nd and 3rd latter: water sample (W), stem surface (SS), whole stem segment (SB); 1st number: day (time); 2nd number: replicate.

normally removed during bacterial community analyses but can be very useful when analyzed separately since they are labile and therefore represent changes in plant cell viability [25].

The relative abundance of chloroplast and mitochondria 16S sequences in stem segments decreased after 3 days in vase water (Fig 1). This could be induced by environmental stress or senescence of cut flowers during the vase period: our results are consistent with the microscopic observations of Simeonova et al. (2000) who found deterioration of chloroplast ultrastructure with swelled and degraded thylakoids and condensation of chromatin in cell nuclei in the senesced yellow-leaf protoplast [26]. This involves programmed cell death (PCD) and is genetically regulated, leading to degradation of nuclear DNA and mitochondria destruction [26].

In addition to senescence, PCD can be stimulated by severe stress and as a reaction to invading microorganisms [27]. Kretschmer et al. (2020) mentioned that bacterial effector proteins from *Pseudomonas syringae*, *Ralstonia solanacearum*, *Candidatus Liberibacter* asiaticus, and *Pantoea stewartia* target chloroplast proteins to affect defense responses and cause proteasomal degradation [28].

We previously visualized numerous bacteria present in xylem vessels of lily stems after 1–2 days in tap water [23], and it is possible that some of these bacteria in the stems are pathogenic and may induce programmed cell death. However, this hypothesis needs further investigation

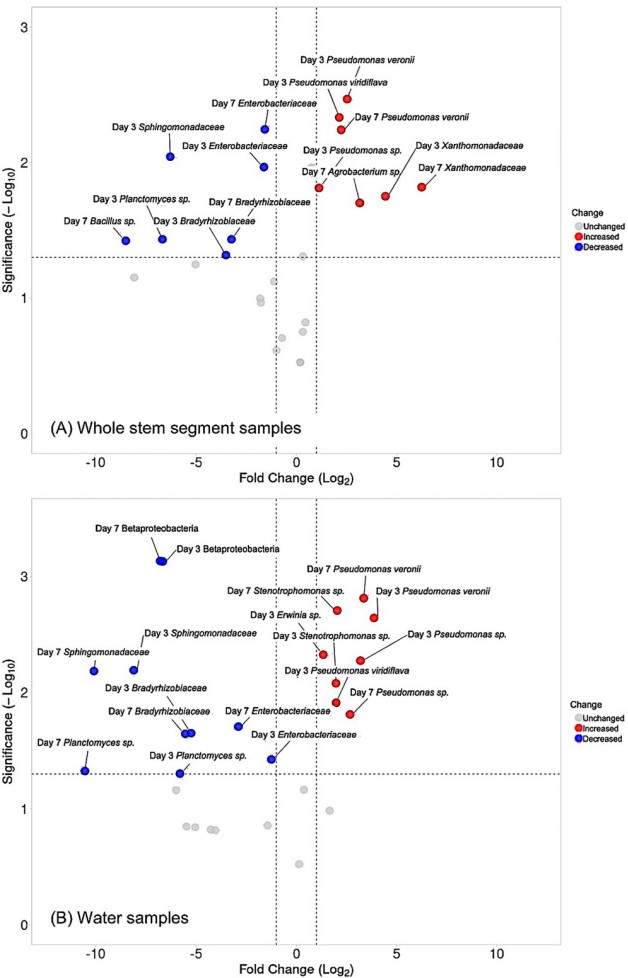

**Fig 8. Volcano plots of top 20 OTUs bacterial taxa of *Gerbera jamesonii*.** (A) Whole stem segment samples (B) Water samples. The Log 2 of fold changes (FC) of bacterial abundance on day 3 and day 7 as compared to day 1. The significance of fold changes was analyzed by using t-test and presented as -Log$_{10}$ (*p*-value) with Bonferroni correction. The threshold of fold change (Log $_2$) is -1 and 1. The threshold of significance (-Log $_{10}$) is 1.3.

and resolution beyond the taxonomic level reported here, as closely related members of the same species can vary dramatically in their pathogenicity.

While we did not study the mechanism behind senescence or PCD, this is the first report to our knowledge, of changes in postharvest 16S amplicon relative abundance from chloroplasts and mitochondria in the stems of cut flowers and may prove useful for future analyses.

The decrease in chloroplast and mitochondria amplicons over time correlated with an increase bacterial amplicons. The increased diversity of bacterial 16S in water and flower stems over time (Fig 2) is likely due in part to the growth of bacteria on flower exudates in water. Our previous experiments also showed that bacterial numbers in vase water containing cut lily stems increased over time [23]. Those observations are consistent with this analysis, showing that time accounted for 1.4 to 3.9-fold more variation than "sample type" (Table 2).

Flower source also appeared to have an effect on bacterial composition although a larger sample size that includes the same varieties from different suppliers is needed to confirm the relevance of this initial observation (Fig 1, Table 1, S1 Table). There are many possible

contributors to the source of microbiome variation including differences in the water and soil used to cultivate the flowers. Vorholt (2012) summarized the origins of microbiota on phyllosphere, suggesting that it includes bacteria distributed by air, water droplets, dust particles, random insect visitors, and local reservoirs, such as soil and pollinators [29].

From the perspective of postharvest logistics, movement of flowers from field to market is complicated and involves multiple steps. In our case, the flowers from the local retailer were shipped and stored in a cooler for more than 5 days after harvest. In contrast, *Lilium* 'Sorbonne' were harvested from a Cornell greenhouse the day the experiment began. Given that the complete production and postharvest history of commercially available flowers is difficult to obtain, future studies should control for flower source as in important variable.

In addition to sources of flowers, our data suggests that flower species had an effect on bacterial communities. *Gerbera's* microbiome was distinct from other the flowers obtained from the same vendor (Fig 3 and Table 1). An obvious characteristic of *Gerbera* is tiny hairs (trichomes) on the stem, whereas rose and lily have smooth stem surfaces. Aleklett et al. (2014) and Vorholt (2012) emphasized that morphologically diverse surface structures on plants provide unique microscale habitats for bacterial colonization. Bacterial community composition may also be affected by plant exudates or volatile compounds [29–31]. Therefore, additional research is needed to understand the relative contributions of plant species, production source, and post-harvest handling on bacterial community composition.

Our work provides important insights into bacterial succession in vase water and on/in the flower stem. In Figs 6 and 7, the heatmaps indicated that the dominant bacteria differed between sample types (water vs stem) and that bacterial composition changed over time, suggestive of ecological succession. For example, *Bacillus spp.*, which were previously reported in cut *Zinnia* vase water [14] and the scape of *Gerbera* [7], were in higher relative abundance on stem samples than water samples on days 0 and 1 of *Gerbera* as compared to days 3 and 7 (Fig 8).

More than half of the OTUs that increased significantly over time in both stem and water samples from *Gerbera* were *Pseudomonas spp.* Members of this genus have frequently been found in studies of cut flower vase water, such as *Gerbera* [12], *Rosa* [22], *Zinnia* [14], *Dianthus* [22], etc. Unlike *Bacillus*, most of the *Pseudomonas* species (Fig 8) that varied with time were detected in both water and stem. *P. veronii*, has previously been associated with soft rot of calla lily [32]. We found that it decreased over time in lily 'Sorbonne' samples but increased over time in *Gerbera* (Fig 8). In addition, we also found that uncharacterized *Xanthomonadaceae* OTUs increased in relative abundance over time in *Gerbera*, but not lily (Figs 7 and 8). This family is well known for its phytopathogenic genera, including *Xanthomonas* and *Xylela* [33]. Another potentially phytopathogenic OTU we detected in *Gerbera* belonged to *Erwinia spp.*, some of which are known for inducing soft rot [34].

As shown in S2 Table, the type of bacteria previously reported from culture dependent analyses, varied widely between studies. Direct comparisons with previous reports is therefore of limited value since most of them focused on vase water at the end of variable vase periods [12, 14, 22, 24]. Our more frequent sampling of both vase water and stems during the vase period demonstrates that sampling time and location can have a significant impact on the community composition. Correlations between the presence of specific OTUs and flower physiology, however, requires additional study. Fang et al. (2021) used hydrogen-rich water as vase solution for cut roses and analyzed the bacterial community at stem-end on day 6 via 16S RNA sequencing. They found that cut roses in hydrogen-rich water (HRW) had longer vase life, and dominant bacteria *Pseudomonas fluorescens* and *Brevundimonas diminuta* in (HRW) were related to vase life enhancement [16]. However, Robinson et al. (2007) had the opposite results that adding

Pseudomonas fluorescence did not prolong cut roses longevity; instead, increasing bacterial concentration decreased total water uptake and vase lives [35].

Follow-on work is needed to help resolve conflicting hypotheses about the relative importance of individual bacterial species versus overall community composition. Carlson et al. (2015) emphasized the primary effect of specific bacteria species, not the overall population, on vase life [14]. Van Doorn et al. (1995), on the other hand, suggested that it is bacterial populations, rather than the presence of specific strains that affect vase life and water uptake of cut carnations [22]. From our research results, we noticed the bacterial communities' succession during the vase period, so the dominant bacteria are dynamic. We hypothesize that the probiotics from the specific strains improve the postharvest quality of cut flowers, and the bacterial concentration plays a key role in the water balance of cut flowers. We need further research to support the hypotheses.

In conclusion, the bacterial diversity varied by flower source, type, and sampling location, and we presented the bacterial succession in flower stems and vase water over time. This would be a critical foundation for further research on postharvest physiology and microbiology. Our use of the V4 region of the 16S rRNA gene to characterize bacterial communities, although convenient and widely used, hampered our ability to ascribe the presence of specific OTUs to pathogenic strains due to lack of taxonomic resolution [36, 37]. Further resolution of the strain versus population question will require a combination of isolate genome sequencing, metagenomics, and the application of Koch's postulates to specific isolates [38, 39]. Further understanding is also needed to determine which steps of postharvest handling affect bacterial populations the most and if those effects play a deterministic role in flower physiology. Research aimed at addressing these and other questions will help us better understand the role of bacteria in postharvest physiology and lead to better postharvest handling of cut flowers.

## Supporting information

**S1 Fig. Shannon entropy, Chao 1, and Faith's phylogenetic diversity of bacterial communities over time in *Rosa* spp. and *Lilium* -As.**
(PDF)

**S1 Dataset.**
(ZIP)

**S1 Table. The comparison of principal components in microbial communities from different flower sources.**
(PDF)

**S2 Table. Identified bacteria in postharvest research of cut flowers.**
(PDF)

## Acknowledgments

Thanks to Jae Won Lyu, Christopher DeRito, and Rose Harmon for helpful insights about experimental methods and data analysis.

## Author Contributions

**Conceptualization:** Yen-Hua Chen, William B. Miller, Anthony Hay.

**Data curation:** Yen-Hua Chen, Anthony Hay.

**Formal analysis:** Yen-Hua Chen.

**Funding acquisition:** William B. Miller.

**Investigation:** Yen-Hua Chen.

**Methodology:** Yen-Hua Chen, Anthony Hay.

**Project administration:** William B. Miller.

**Resources:** Anthony Hay.

**Software:** Anthony Hay.

**Supervision:** William B. Miller, Anthony Hay.

**Validation:** Anthony Hay.

**Visualization:** Yen-Hua Chen.

**Writing – original draft:** Yen-Hua Chen.

**Writing – review & editing:** Yen-Hua Chen, William B. Miller, Anthony Hay.

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
