## [Decision Letter · Decision Letter 0]

21 Jul 2023

PONE-D-23-15976Microbial succession on cut flowersPLOS ONE

Dear Dr. Chen,

Thank you for submitting your manuscript to PLOS ONE. After careful consideration, we feel that it has merit but does not fully meet PLOS ONE’s publication criteria as it currently stands. Therefore, we invite you to submit a revised version of the manuscript that addresses the points raised during the review process.

We look forward to receiving your revised manuscript.

Kind regards,

Mojtaba Kordrostami, Ph.D.

Academic Editor

PLOS ONE

Journal Requirements:

3. We notice that your supplementary [figures/tables] are included in the manuscript file. Please remove them and upload them with the file type 'Supporting Information'. Please ensure that each Supporting Information file has a legend listed in the manuscript after the references list.

Additional Editor Comments :

Dear Authors,

We have now completed our review of your manuscript entitled "Microbial succession on cut flowers". I want to express my gratitude for your patience during this review process.

Two expert reviewers have provided thorough assessments of your manuscript. While the study has been seen as potentially valuable, offering insights into the dynamic succession of microbial communities in cut flower vase ecosystems, both reviewers agree that significant revisions are necessary before it can be accepted for publication in PLOS ONE.

Reviewer 1 suggested changes to the title to better reflect the content of the study, recommended specifying the sampling times, and requested improvements in the results section for clarity. They also found discrepancies in the data provided in the text and figures, and queried about the SRA accession numbers for the samples. There were also concerns raised about the choice of Gerbera jamesonii for specific OTUs over time analysis. They pointed out that Latin names of bacteria in figures need to be italicized, and insisted on consistency in the reference format and use of terms like "Figure" or "Fig" in the manuscript.

Reviewer 2 noted lack of clarity in the objectives of your study, with suggestions to enhance the introduction by including more background information and citations. They also emphasized the need for consistency in the use of terms such as "microbes," "bacteria," and "microbial communities." Reviewer 2 recommends a more comprehensive discussion of your results, including any limitations or potential sources of bias in your study, and the addition of a separate conclusion section.

We invite you to revise your manuscript to address these comments. In your response, please provide a point-by-point response to the issues raised, indicating where changes have been made in the manuscript or why no change was deemed necessary. If you are able to make these revisions, we would be glad to reconsider your manuscript for publication in PLOS ONE.

Please note that this does not guarantee your manuscript will be accepted for publication. Your revised manuscript will be sent back to the reviewers for further evaluation and comments.

We look forward to receiving your revised manuscript.

Best Regards,

Mojtaba Kordrostami

Editor,

PLOS ONE

Reviewers' comments:

Reviewer's Responses to Questions

**Comments to the Author**

1. Is the manuscript technically sound, and do the data support the conclusions?

Reviewer #1: Partly

Reviewer #2: Yes

2. Has the statistical analysis been performed appropriately and rigorously? 

Reviewer #1: Yes

Reviewer #2: Yes

3. Have the authors made all data underlying the findings in their manuscript fully available?

Reviewer #1: Yes

Reviewer #2: Yes

4. Is the manuscript presented in an intelligible fashion and written in standard English?

Reviewer #1: Yes

Reviewer #2: Yes

5. Review Comments to the Author

Reviewer #1: The authors studied the dynamic succession of microbial communities in the vase ecosystem of different cut flower varieties at the 16S rRNA gene level, providing clues for managing cut flowers as well as improving the longevity of cut flower vases. However, there are some issues in manuscript that need to be further addressed, as follows:

1.The authors study the succession of microbial communities in cut flower vase water, which I think should be presented in the title. However, the title “Microbial succession on cut flowers” does not reflect the study fully.

2.“…each vase holding flowers on day 1, 3, and 7, and the control vase on day 0, 1, 3, 7.” Sampling time should be included here.

3.In the results section, it is suggested that the subtitles should be added for descriptions.

4.Line 162 of the manuscript mentions that "In all 4 flower types, the initial relative abundance of chloroplast sequences ranged from 69-82% of all amplicons." However, Fig 1a shows that the initial relative abundance of chloroplast in Lily-As ranges from 20-30%, which is not consistent with the text description. Please check the results section for the same mistakes.

5.I couldn’t see the SRA (sequence read archive) accession numbers for the samples.

6.The authors analyzed stem and water samples of Gerbera jamesonii in order to better understand changes in specific OTUs over time. For this reason, I wonder why Gerbera jamesonii was chosen? Can gerbera daisies represent the remaining three flowers?

7.The Latin name of the bacterium should be italicized as it appears in Figures 6, 7, and 8.

8.Please keep the reference format consistent in the manuscript, and “Figure” or “Fig”, please use the same style in manuscript.

Reviewer #2: Dear authors

Overall, the this paper provides a comprehensive overview of a study on microbial succession on cut flowers and its impact on post harvest quality. However, there are some weaknesses in the text that can be addressed:

1- Lack of clarity in the objectives: The introduction section does not clearly state the specific objectives of the study. It should clearly define the research questions or hypotheses that the study aimed to address.

2- Inadequate background information: The text lacks a thorough literature review to establish the existing knowledge and research gaps in the field. Including more background information and citing relevant studies would strengthen the introduction.

3- Inconsistent use of terminology: The text inconsistently uses terms such as "microbes," "bacteria," and "microbial communities." Standardizing the terminology would improve the clarity and precision of the text.

4- Inadequate discussion of results: The discussion section should provide a more in-depth analysis and interpretation of the results, linking them back to the research objectives and existing literature. It should also address any limitations or potential sources of bias in the study.

5-Lack of conclusion: The text does not include a separate conclusion section summarizing the main findings of the study. Adding a concise conclusion would provide a clear ending to the research article.

By addressing these weaknesses, the text would become more coherent, informative, and engaging for readers.

Best Regards

6. PLOS authors have the option to publish the peer review history of their article (what does this mean?). If published, this will include your full peer review and any attached files.

Reviewer #1: No

Reviewer #2: **Yes: **Ali Akbar Ghasemi-Soloklui

---

## [Author Response · Author response to Decision Letter 0]

20 Sep 2023

Dear reviewers and editors,

I have revised the manuscript based on your comments. 

The major revisions are:

1. Revised manuscript according to reviewers' and editors’ comments.

2. Manuscript format edit, upload figures and supporting information.

3. Submitted SRA in NCBI (BioProject ID: PRJNA1008676).

4. Uploaded the minimal data set in supporting files.

Please check the rebuttal letter for more explanation, and the Line# mentioned in the rebuttal letter is according to the "Review-All Markup" situation. Thank you.

---

## [Decision Letter · Decision Letter 1]

25 Sep 2023

Postharvest bacterial succession on cut flowers and vase water

PONE-D-23-15976R1

Dear Dr. Chen,

We’re pleased to inform you that your manuscript has been judged scientifically suitable for publication and will be formally accepted for publication once it meets all outstanding technical requirements.

Kind regards,

Mojtaba Kordrostami, Ph.D.

Academic Editor

PLOS ONE

Additional Editor Comments (optional):

The manuscript can be accepted now.

Reviewers' comments:

Reviewer's Responses to Questions

**Comments to the Author**

1. If the authors have adequately addressed your comments raised in a previous round of review and you feel that this manuscript is now acceptable for publication, you may indicate that here to bypass the “Comments to the Author” section, enter your conflict of interest statement in the “Confidential to Editor” section, and submit your "Accept" recommendation.

Reviewer #1: All comments have been addressed

2. Is the manuscript technically sound, and do the data support the conclusions?

Reviewer #1: (No Response)

3. Has the statistical analysis been performed appropriately and rigorously? 

Reviewer #1: (No Response)

4. Have the authors made all data underlying the findings in their manuscript fully available?

Reviewer #1: (No Response)

5. Is the manuscript presented in an intelligible fashion and written in standard English?

Reviewer #1: (No Response)

6. Review Comments to the Author

Reviewer #1: (No Response)

7. PLOS authors have the option to publish the peer review history of their article (what does this mean?). If published, this will include your full peer review and any attached files.

Reviewer #1: No

---

## [Editor Report · Acceptance letter]

29 Sep 2023

PONE-D-23-15976R1 

Postharvest bacterial succession on cut flowers and vase water 

Dear Dr. Chen:

I'm pleased to inform you that your manuscript has been deemed suitable for publication in PLOS ONE. Congratulations! Your manuscript is now with our production department. 

Kind regards, 

on behalf of

Dr. Mojtaba Kordrostami 

Academic Editor

PLOS ONE